# Progress toward closing gaps in the hepatitis C virus cascade of care for people who inject drugs in San Francisco

Ali Mirzazadeh[1,2]*, Yea-Hung Chen[1], Jess Lin[3], Katie Burk[4], Erin C. Wilson[3], Desmond Miller[3], Danielle Veloso[3], Willi McFarland[2,3], Meghan D. Morris[1]

**1** Department of Epidemiology and Biostatistics, University of California, San Francisco, California, United States of America, **2** Institute for Global Health Sciences, University of California, San Francisco, California, United States of America, **3** Center for Public Health Research, San Francisco Department of Public Health, San Francisco, California, United States of America, **4** Community Health Equity and Promotion Branch, San Francisco Department of Public Health, San Francisco, California, United States of America

\* Ali.Mirzazadeh@ucsf.edu

## Abstract

### Background

People who inject drugs (PWID) are disproportionately affected by hepatitis C virus (HCV). Data tracking the engagement of PWID in the continuum of HCV care are needed to assess the reach, target the response, and gauge impact of HCV elimination efforts.

### Methods

We analyzed data from the National HIV Behavioral Surveillance (NHBS) surveys of PWID recruited via respondent driven sampling (RDS) in San Francisco in 2018. We calculated the number and proportion who self-reported ever: (1) tested for HCV, (2) tested positive for HCV antibody, (3) diagnosed with HCV, (4) received HCV treatment, (5) and attained sustained viral response (SVR). To assess temporal changes, we compared 2018 estimates to those from the 2015 NHBS sample.

### Results

Of 456 PWID interviewed in 2018, 88% had previously been tested for HCV, 63% tested antibody positive, and 50% were diagnosed with HCV infection. Of those diagnosed, 42% received treatment. Eighty-one percent of those who received treatment attained SVR. In 2015 a similar proportion of PWID were tested and received an HCV diagnosis, compared to 2018. However, HCV treatment was more prevalent in the 2018 sample (19% vs. 42%, P-value 0.01). Adjusted analysis of 2018 survey data showed having no health insurance (APR 1.6, P-value 0.01) and having no usual source of health care (APR 1.5, P-value 0.01) were significantly associated with untreated HCV prevalence.

**Data Availability Statement:** The data that we used were collected as part of the National HIV Behavioral Surveillance (NHBS). The NHBS is to conduct behavioral surveillance among persons at

high rics for HIV infection, including people who inject drugs. Due to sensitivity of the data and vulnerability of the target populations, researchers seeking individual-level data may apply to analyze NHBS data for San Francisco by contacting the Center for Public Health Research at Sean.Arayasirikul@sfdph.org.

**Funding:** The Centers for Disease Control and Prevention (CDC) funded the collection of data for the National HIV Behavioral Surveillance 2015 and 2018 round (WM supported by grant number 5U1BPS003247). The funder had no role in the analysis or interpretation of data, the writing of the report, or the decision to submit the manuscript for publication. AM was supported by National Institute on Drug Abuse (grant number 5R37DA015612-170 and MDM by National Institute on Drug Abuse (grant number R21DA046809).

**Competing interests:** NO authors have competing interests.

## Conclusion

While findings indicate an improvement in HCV treatment uptake among PWID in San Francisco, more than half of PWID diagnosed with HCV infection had not received HCV treatment in 2018. Policies and interventions to increase coverage are necessary, particularly among PWID who are uninsured and outside of regular care.

## Introduction

In the United States, more than 4.1 million people were infected with hepatitis C virus (HCV) during 2013–2016, of whom 2.4 million persons were viremic (HCV RNA-positive) [1]. Annual mortality of HCV was 4.13 per 100,00 population in 2017 [2], greater than mortality due to HIV (1.7 per 100,00 population) [3].

The majority of HCV transmission (80%) occurs among people who inject drugs (PWID) [4]. Of an estimated 22,000 San Francisco who were positive for HCV antibodies (anti-HCV positive), indicating exposure to HCV, in 2016 [5], 66% were PWID. Moreover, few (23%) PWID had received HCV treatment [5]. In 2011, new direct-acting antiviral (DAA) HCV therapies offered well-tolerated treatment that can cure more than 95% of those who complete a 8–12 week treatment course [6,7]. However, the few studies of DAA among PWID suggest few (10–23%) PWID had received HCV treatment [5,8]. Expanding DAA treatment among PWID is a key step to achieving the 2030 HCV elimination goals set by the U.S. and World Health Organization (WHO) [9,10].

The HCV Continuum of Care (CoC) describes successive stages from disease identification to cure and can be used to monitor population-level public health outcomes, identify gaps in testing and treatment, and offer insights into opportunities for intervention. We leveraged data from the National Health and Behavioral Surveillance (NHBS), a tri-annual community-based survey of PWID in San Francisco, to characterize the HCV CoC. Key indicators measured included HCV testing, antibody positivity, diagnosis of infection, treatment, and sustained virologic response (SVR). We assessed temporal changes in the HCV care cascade by comparing results from 2018 to the 2015 NHBS. Lastly, we identified correlates of those who were diagnosed with HCV infection but had not started HCV treatment.

## Methods

Study data came from the 2015 and 2018 cycles of cross-sectional surveys of PWID collected by the San Francisco Department of Public Health as part of the CDC's National HIV Behavioral Surveillance system (NHBS), a multi-site biobehavioral surveillance system [11]. Participants were recruited though respondent-driven sampling (RDS), a sampling and recruitment method based on long-chain peer referrals aimed at attaining a diverse representation of PWID. Complete NHBS methods have been previously described [12].

Eligibility criteria for both cycles were: a) 18 years of age or older, b) reported injection of illicit drugs in the past 12 months, c) had been recruited via RDS methods (i.e., were an initial study seed or given a study coupon by another participant), and d) a resident of San Francisco or San Mateo county. Upon informed consent, participants completed an interviewer-administered questionnaire on demographics, social network characteristics, recent and lifetime drug use behaviors, drug treatment, and medical service access (HIV, HCV, mental health). Participants also underwent a blood draw to assess HCV and HIV infection. Study procedures

were conducted anonymously, and participants were not linked across cross-sectional survey waves. Enrolled participants were offered $75 for their participation. If participants successfully referred someone into the study, they were given $10 for each eligible referral. The study protocol as reviewed and approved by the University of California, San Francisco's IRB (IRB code 17–21489). The IRB approved the payment (remuneration), recognizing that the form and amount of payment is consistent with expectations of participants in similar studies. We did not follow-up with participants as the study was cross-sectional and to preserve confidentiality no names or contact information was asked for or collected.

## Data analysis

Only self-reported information was used to assess HCV care cascade milestones. First, we calculated the number and proportion of participants across the following HCV care cascade steps: (1) tested for HCV, (2) tested positive for HCV antibody, (3) diagnosed with HCV infection, (4) received HCV treatment, (5) and attained SVR. The timeframe for our analysis was "ever". Our analysis of cascade for HCV testing, diagnosis and treatment was based on the following survey questions:

- Have you ever been tested for hepatitis C infection?

- Has a doctor, nurse or other health care provider ever told you that you had hepatitis C?

- Have you ever taken medicine to treat your hepatitis C infection?

The denominators for percent who "tested for HCV", "tested positive for HCV antibody", and "diagnosed with HCV infection" were the number of recruited people in the study. The denominators for percent who "received treatment" and "attained SVR" were those diagnosed with HCV infection. Participants with missing response data for a particular cascade stage were excluded from that cascade stage's numerator and subsequent cascade stage's denominator.

We reported both crude and RDS-adjusted estimates for five HCV care cascade steps. For crude estimates, we calculated the exact Poisson confidence intervals [13] assuming simple random sampling. For RDS-adjusted estimates which considered network size and homophily within networks, we used RDS package in R software [14] and Gile's successive sampling estimator [15] to produce weighted prevalence estimates for HCV care cascade measures. Because "tested positive for antibody" and "attained SVR" were not measured in the 2015 survey, we were not able to report these measures for 2015.

We also measured HCV treatment prevalence overall and among subgroups of self-reported HCV-infected PWID in 2018 by selected sociodemographic characteristics: age, race/ethnicity, gender, education, employment, sexual identity, homeless, health insurance, usual source of health care, had visited a physician in last 12 months, age at first injection, and injection frequency. Homeless was defined as living on the street, in a shelter, in a Single Room Occupancy hotel (SRO), or in a car. We also reported the percent of untreated HCV among subgroups of PWID with a history of self-reported HCV diagnosis in 2015 and 2018. We calculated the differences in percent untreated for 2015 and 2018 subgroups and tested for significant differences by Chi square or Fisher exact test. We then used Poisson models with log links and robust standard errors to evaluate the association between above listed variables and remaining untreated. The crude and adjusted prevalence ratios (APR) were reported as measure of association. For each variable included in the multivariable Poisson analysis, we controlled for age, race/ethnicity, and gender. We reported the results from unweighted model based on a simulation study that showed unweighted regression models perform better than weighted regression techniques for RDS data [16]. Analyses were done in R software [17].

## Results

In 2018 (**Table 1**), 456 PWID were recruited to the study. The majority were older than 40 years of age (60.3%), White (46.3%), male (67.6%), high-school graduates (73.9%), currently unemployed (45.2%), heterosexual (74.0%), currently homeless (77.2%), insured for medical care (93.6%), and reported a usual source of health care (87.5%). The majority of PWID in 2018 reported visiting a physician in the last 12 months (87.1%), started injecting drugs before age 25 years old (66.7%), and injected more than once per week (89.7%). Compared to the 2018 sample, a greater proportion of participants in the 2015 sample were Black (26.6% in 2015 vs. 19.1% in 2018) and had accessed any drug treatment in the last 12 months (41.3% vs. 28.7%). A lesser proportion of participants in 2015 were currently homeless (68.2% vs. 77.2%), insured for medical care (87.8% vs. 93.6%), and had accessed a needle exchange in the prior year (22.8% vs. 91.2%).

The RDS-adjusted estimates for the HCV infection treatment cascade are presented in Fig 1. Among PWID in 2018, 88% had ever been tested for HCV, 63% had tested HCV antibody positive, and 50% had received an HCV infection diagnosis. Of those diagnosed, 42% reported receiving HCV treatment, and 34% reported attaining SVR. While a similar proportion of the PWID population in 2015 had ever been tested for HCV (87%) and diagnosed with HCV infection (47%), a smaller proportion (19% vs. 42%, P-value 0.01) had ever received HCV treatment. Looking at crude (not RDS-adjusted) estimates for the HCV treatment cascade, similar progress was observed (S1 Fig).

When controlling for age, race/ethnicity and gender, we found some differences in independent associations of untreated HCV infection in 2015 and 2018. Among the 2015 sample, identifying as transgender (APR 1.2, P-value 0.01), having no usual source of health care (APR 1.1, P-value 0.04), and starting injection drug use between ages 35–44 years (APR 1.3, P-value 0.03), were positively associated with untreated HCV infection prevalence. For the 2018 sample, having no health insurance (APR 1.6, P-value 0.01) and having no usual source of health care (APR 1.5, P-value 0.01) were positively associated with untreated HCV infection prevalence (**Table 2**).

When comparing the prevalence of untreated HCV infection among subgroups in 2015 to subgroups in 2018, we found that several subgroups experienced a significant decrease in untreated HCV prevalence (**Table 2**). These subgroups were: PWID over the age of 30 years (-21.9 to -43.6%, p-value<0.011) within all race/ethnicity groups (-21.9% to -39.4%, p-value < 0.021) and all gender groups (-27.2% male and -21.3% female and transgender -100.0%, p-value 0.001), those with high school education or below (-30.7, p-value 0.001), all employment groups, with the largest decrease among those recently employed (-46.7%, p-value 0.001), both homeless and non-homeless PWID (-24.2 and -33.2%, p-value 0.001 respectively), those insured for medical care (-27.1%, p-value 0.001), those reporting a usual source of medical care (-28.8%, p-value 0.001), and those who recently accessed a needle exchange program (-52.7%, p-value 0.001).

## Discussion

Our results indicate a near tripling (15% vs. 41%) between 2015 and 2018 of PWID with diagnosed HCV infection starting HCV treatment in San Francisco. Priority gaps remain, as over half (59%) of PWID reporting an HCV diagnosis in 2018 reported they had not received HCV treatment. PWID who were younger than 30, unemployed, or no usual source of health care had a lower probability of accessing HCV treatment. Remarkably, all ten transgender persons recruited in 2018 were treated for HCV, while in 2015 transgender identity was significantly associated with untreated HCV. While a greater percent of homeless PWID started HCV treatment in 2018 than in 2015, a sustained effort is needed to reach the almost 70% of homeless PWID with diagnosed HCV infection who remain without HCV treatment.

**Table 1. Select characteristics of people who inject drugs participating in the 2015 and 2018 National HIV Behavioral Surveillance surveys, San Francisco.**

| Variable | 2015 | 2018 | P-Value |
|---|---|---|---|
| Total | 528 (100.0) | 456 (100.0) | |
| Age | | | |
| 18–29 | 68 (12.9) | 51 (11.2) | 0.88 |
| 30–39 | 117 (22.2) | 109 (23.9) | |
| 40–49 | 126 (23.9) | 109 (23.9) | |
| 50–64 | 196 (37.1) | 166 (36.4) | |
| 65+ | | | |
| Race/ethnicity | | | |
| Black | 140 (26.6) | 86 (19.1) | 0.01 |
| Hispanic | 69 (13.1) | 69 (15.3) | |
| White | 254 (48.3) | 209 (46.3) | |
| Other | 18 (3.4) | 20 (4.4) | |
| Mixed | 45 (8.6) | 67 (14.9) | |
| Gender | | | |
| Male | 381 (72.2) | 305 (67.6) | 0.18 |
| Female | 141 (26.7) | 136 (30.2) | |
| Transgender | 6 (1.1) | 10 (2.2) | |
| Education | | | |
| No high-school degree | 123 (23.3) | 92 (20.2) | 0.49 |
| High-school degree | 376 (71.2) | 337 (73.9) | |
| College degree or beyond | 29 (5.5) | 27 (5.9) | |
| Employment | | | |
| Employed | 44 (8.3) | 43 (9.4) | 0.91 |
| Unemployed | 236 (44.7) | 206 (45.2) | |
| Unable to work | 203 (38.4) | 171 (37.5) | |
| Other | 45 (8.5) | 36 (7.9) | |
| Sexual identity | | | |
| Heterosexual | 356 (67.8) | 330 (74.0) | 0.08 |
| Homosexual | 53 (10.1) | 31 (7.0) | |
| Bisexual | 116 (22.1) | 85 (19.1) | |
| Homeless (current) | | | |
| Yes | 360 (68.2) | 352 (77.2) | 0.01 |
| No | 168 (31.8) | 104 (22.8) | |
| Health insurance (current) | | | |
| Yes | 460 (87.8) | 422 (93.6) | 0.01 |
| No | 64 (12.2) | 29 (6.4) | |
| Usual source of care | | | |
| Yes | 445 (84.3) | 399 (87.5) | 0.18 |
| No | 83 (15.7) | 57 (12.5) | |
| Accessed health care, last 12 months | | | |
| Yes | 458 (86.7) | 397 (87.1) | 0.96 |
| No | 70 (13.3) | 59 (12.9) | |
| Age at first injection | | | |
| 0–12 | 24 (4.5) | 18 (4.0) | 0.90 |
| 13–17 | 150 (28.4) | 130 (28.6) | |
| 18–24 | 181 (34.3) | 155 (34.1) | |

*(Continued)*

**Table 1.** (Continued)

| Variable | 2015 | 2018 | P-Value |
|---|---|---|---|
| 25–34 | 113 (21.4) | 99 (21.8) | |
| 35–44 | 43 (8.1) | 43 (9.5) | |
| 45+ | 17 (3.2) | 10 (2.2) | |
| Injection more than once per week | | | |
| Yes | 482 (91.3) | 409 (89.7) | 0.46 |
| No | 46 (8.7) | 47 (10.3) | |
| Accessed needle exchange, last 12 months | | | |
| Yes | 120 (22.8) | 416 (91.2) | 0.01 |
| No | 406 (77.2) | 40 (8.8) | |
| Drug treatment, last 12 months | | | |
| Yes | 218 (41.3) | 131 (28.7) | 0.01 |
| No | 310 (58.7) | 325 (71.3) | |
| HIV status | | | |
| Positive | 62 (11.8) | 42 (9.3) | 0.25 |
| Negative | 464 (88.2) | 410 (90.7) | |

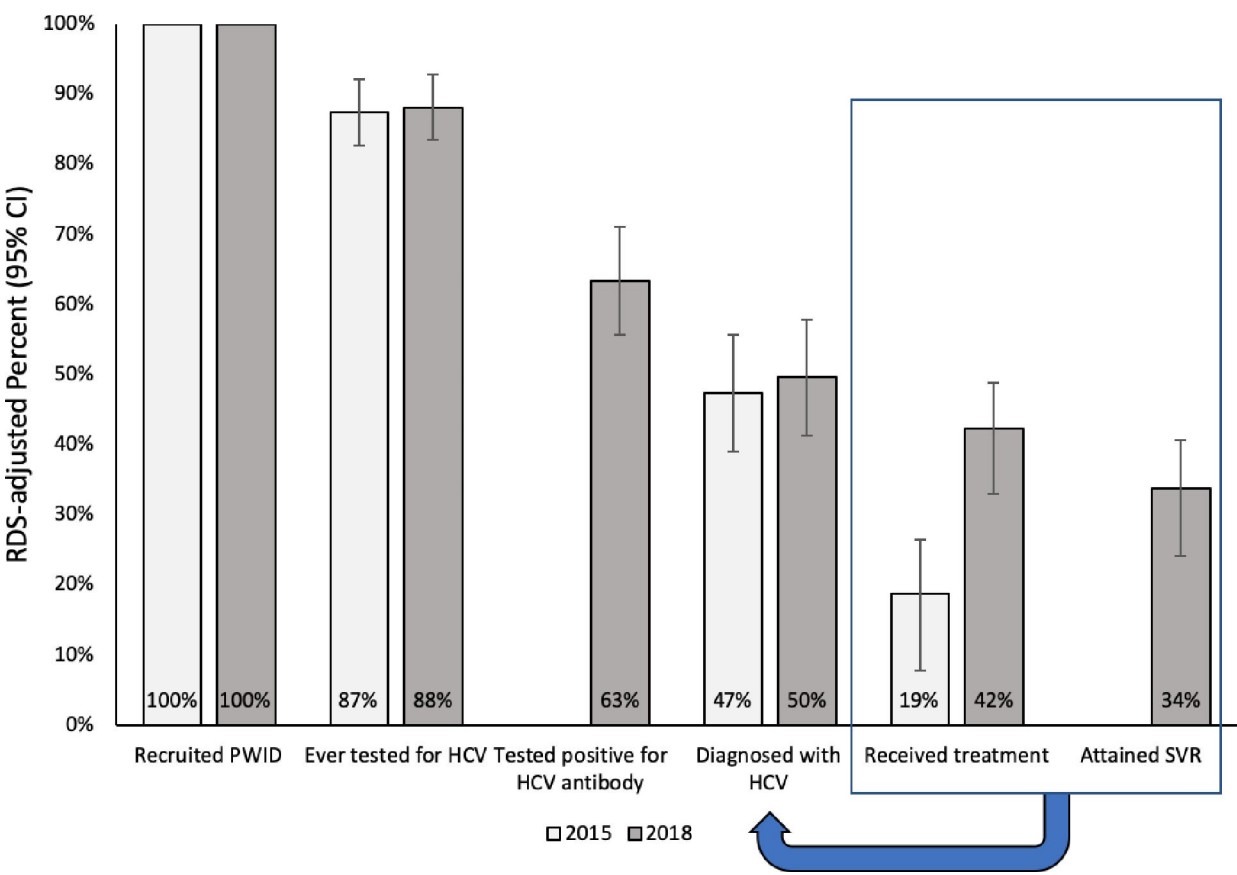

**Fig 1. RDS-adjusted estimates for the Hepatitis C virus (HCV) infection treatment cascade in two rounds of surveys of people who inject drugs (PWID) in San Francisco, 2015 and 2018.** Denominators for percent who received treatment and attained sustained virologic response (SVR) are those diagnosed with HCV infection. In 2015, tested antibody positive and attained SVR were not measured. The estimates in this graph are RDS-adjusted.

**Table 2. Unadjusted and adjusted factors associated with prevalence of untreated HCV infection among PWID who self-reported ever receiving an HCV diagnosis, 2015 and 2018.**

| Variables | 2015 | | | | | 2018 | | | | | 2015 vs. 2018 | |
|---|---|---|---|---|---|---|---|---|---|---|---|---|
| | % untreated HCV | Crude | | Adjusted | | % untreated HCV | Crude | | Adjusted | | Diff. in % untreated | P-value |
| | | PR (95% CI) | P-value | PR (95% CI) | P-value | | PR (95% CI) | P-value | PR (95% CI) | P-value | | |
| **Age** | | | | | | | | | | | | |
| 18–29 | 93.8 | 1 | 1 | 1 | 1 | 84.0 | 1 | | 1 | | -9.8 | 0.081 |
| 30–39 | 85.7 | 0.9 (0.8, 1.1) | 0.23 | 0.9 (0.8, 1.1) | 0.27 | 63.8 | 0.8 (0.6, 1.0) | 0.05 | 0.7 (0.5, 1.0) | 0.03 | -21.9 | 0.001 |
| 40–49 | 91.3 | 1.0 (0.9, 1.1) | 0.65 | 1.0 (0.9, 1.1) | 0.89 | 55.8 | 0.7 (0.5, 0.9) | 0.01 | 0.6 (0.5, 0.9) | 0.01 | -35.5 | 0.001 |
| 50–64 | 80.8 | 0.9 (0.8, 1.0) | 0.02 | 0.9 (0.8, 1.0) | 0.06 | 55.7 | 0.7 (0.5, 0.9) | 0.01 | 0.6 (0.5, 0.8) | 0.01 | -25.1 | 0.001 |
| 65+ | 72.2 | 0.8 (0.6, 1.0) | 0.09 | 0.8 (0.6, 1.1) | 0.12 | 28.6 | 0.3 (0.1, 0.8) | 0.01 | 0.3 (0.1, 0.8) | 0.01 | -43.6 | 0.011 |
| **Race/ethnicity** | | | | | | | | | | | | |
| White | 88.6 | 1 | 1 | 1 | 1 | 62.9 | 1 | | 1 | | -25.7 | 0.001 |
| Black | 78.2 | 0.9 (0.8, 1.0) | 0.06 | 0.9 (0.8, 1.1) | 0.10 | 50.0 | 0.8 (0.6, 1.1) | 0.20 | 0.9 (0.6, 1.3) | 0.55 | -28.2 | 0.001 |
| Hispanic | 84.8 | 1.0 (0.8, 1.1) | 0.52 | 1.0 (0.8, 1.1) | 0.65 | 62.9 | 1.0 (0.7, 1.3) | 0.99 | 1.0 (0.8, 1.4) | 0.83 | -21.9 | 0.001 |
| Other | 72.7 | 0.8 (0.6, 1.2) | 0.29 | 0.8 (0.6, 1.1) | 0.30 | 33.3 | 0.5 (0.2, 1.7) | 0.27 | 0.6 (0.2, 1.6) | 0.27 | -39.4 | 0.021 |
| Mixed | 89.7 | 1.0 (0.9, 1.2) | 0.87 | 1.0 (0.9, 1.2) | 0.83 | 53.3 | 0.8 (0.6, 1.2) | 0.37 | 0.9 (0.6, 1.2) | 0.48 | -36.4 | 0.001 |
| **Gender** | | | | | | | | | | | | |
| Male | 83.6 | 1 | 1 | 1 | 1 | 56.4 | 1 | | 1 | | -27.2 | 0.001 |
| Female | 87.5 | 1.0 (0.9, 1.2) | 0.39 | 1.0 (0.9, 1.1) | 0.55 | 66.2 | 1.2 (0.9, 1.5) | 0.16 | 1.2 (1.0, 1.5) | 0.11 | -21.3 | 0.001 |
| Transgender | 100.0 | 1.2 (1.1, 1.3) | 0.01 | 1.2 (1.1, 1.3) | 0.01 | 0.0 | | | | | -100.0 | 0.001 |
| **Education** | | | | | | | | | | | | |
| No high school | 80.8 | 1 | 1 | 1 | 1 | 69.4 | 1 | | 1 | | -11.4 | 0.061 |
| High school | 86.7 | 1.1 (0.9, 1.2) | 0.26 | 1.1 (0.9, 1.2) | 0.33 | 56.0 | 0.8 (0.6, 1.0) | 0.07 | 0.8 (0.7, 1.0) | 0.13 | -30.7 | 0.001 |
| College or beyond | 78.6 | 1.0 (0.7, 1.3) | 0.85 | 1.0 (0.7, 1.3) | 0.75 | 54.5 | 0.8 (0.4, 1.4) | 0.41 | 0.8 (0.5, 1.5) | 0.53 | -24.1 | 0.061 |
| **Employment** | | | | | | | | | | | | |
| Employed | 86.7 | 1 | 1 | 1 | 1 | 40.0 | 1 | | 1 | | -46.7 | 0.001 |
| Unemployed | 89.3 | 1.0 (0.8, 1.3) | 0.77 | 1.0 (0.8, 1.2) | 0.98 | 65.3 | 1.6 (0.9, 2.8) | 0.08 | 1.6 (0.9, 2.8) | 0.12 | -24.0 | 0.001 |
| Unable to work | 81.2 | 0.9 (0.8, 1.2) | 0.55 | 0.9 (0.8, 1.1) | 0.45 | 55.0 | 1.4 (0.8, 2.4) | 0.27 | 1.4 (0.8, 2.5) | 0.22 | -26.2 | 0.001 |
| Other | 82.1 | 0.9 (0.7, 1.2) | 0.69 | 0.9 (0.7, 1.2) | 0.46 | 72.7 | 1.8 (1.0, 3.5) | 0.07 | 1.8 (0.9, 3.4) | 0.09 | -9.4 | 0.281 |
| **Sexual identity** | | | | | | | | | | | | |
| Heterosexual | 84.7 | 1 | 1 | 1 | 1 | 57.3 | 1 | | 1 | | -27.4 | 0.001 |
| Homosexual | 86.2 | 1.0 (0.9, 1.2) | 0.83 | 1.0 (0.9, 1.2) | 0.58 | 66.7 | 1.2 (0.8, 1.7) | 0.44 | 1.2 (0.8, 1.7) | 0.39 | -19.5 | 0.041 |
| Bisexual | 84.1 | 1.0 (0.9, 1.1) | 0.91 | 1.0 (0.9, 1.1) | 0.78 | 60.5 | 1.1 (0.8, 1.4) | 0.70 | 1.0 (0.7, 1.3) | 0.84 | -23.6 | 0.001 |
| **Homeless (current)** | | | | | | | | | | | | |
| No | 81.4 | 1 | 1 | 1 | 1 | 48.2 | 1 | | 1 | | -33.2 | 0.001 |
| Yes | 86.7 | 1.1 (1.0, 1.2) | 0.25 | 1.0 (0.9, 1.1) | 0.42 | 62.4 | 1.3 (1.0, 1.7) | 0.09 | 1.2 (0.9, 1.7) | 0.17 | -24.3 | 0.001 |
| **Health insurance (current)** | | | | | | | | | | | | |
| Yes | 84.1 | 1 | 1 | 1 | 1 | 57.0 | 1 | | 1 | | -27.1 | 0.001 |
| No | 92.0 | 1.1 (1.0, 1.2) | 0.17 | 1.1 (1.0, 1.2) | 0.23 | 90.0 | 1.6 (1.2, 2.0) | 0.01 | 1.6 (1.3, 2.0) | 0.01 | -2.0 | 0.691 |
| **Usual source of care** | | | | | | | | | | | | |
| Yes | 83.8 | 1 | 1 | 1 | 1 | 55.0 | 1 | | 1 | | -28.8 | 0.001 |
| No | 96.3 | 1.1 (1.0, 1.3) | 0.01 | 1.1 (1.0, 1.2) | 0.04 | 91.7 | 1.7 (1.4, 2.0) | 0.01 | 1.5 (1.2, 1.8) | 0.01 | -4.6 | 0.201 |
| **Accessed health care, last 12 months** | | | | | | | | | | | | |
| Yes | 84.3 | 1 | 1 | 1 | 1 | 56.9 | 1 | | 1 | | -27.4 | 0.001 |
| No | 90.3 | 1.1 (0.9, 1.2) | 0.28 | 1.1 (0.9, 1.2) | 0.39 | 75.0 | 1.3 (1.0, 1.7) | 0.04 | 1.3 (1.0, 1.7) | 0.07 | -15.3 | 0.021 |

*(Continued)*

**Table 2.** (Continued)

| Variables | 2015 | | | | | 2018 | | | | | 2015 vs. 2018 | |
|---|---|---|---|---|---|---|---|---|---|---|---|---|
| | % untreated HCV | Crude | | Adjusted | | % untreated HCV | Crude | | Adjusted | | Diff. in % untreated | P-value |
| | | PR (95% CI) | P-value | PR (95% CI) | P-value | | PR (95% CI) | P-value | PR (95% CI) | P-value | | |
| **Age at first injection** | | | | | | | | | | | | |
| 0–12 | 77.8 | 1 | 1 | 1 | 1 | 69.2 | 1 | | 1 | | -8.6 | 0.361 |
| 13–17 | 83.5 | 1.1 (0.8, 1.4) | 0.59 | 1.1 (0.8, 1.4) | 0.56 | 51.2 | 0.7 (0.5, 1.1) | 0.16 | 0.8 (0.5, 1.3) | 0.36 | -32.3 | 0.001 |
| 18–24 | 85.1 | 1.1 (0.8, 1.4) | 0.49 | 1.1 (0.8, 1.4) | 0.52 | 65.4 | 0.9 (0.6, 1.4) | 0.78 | 1.0 (0.6, 1.5) | 0.99 | -19.7 | 0.001 |
| 25–34 | 85.7 | 1.1 (0.8, 1.4) | 0.48 | 1.1 (0.9, 1.5) | 0.34 | 62.9 | 0.9 (0.6, 1.4) | 0.67 | 1.0 (0.7, 1.7) | 0.81 | -22.8 | 0.001 |
| 35–44 | 100.0 | 1.3 (1.0, 1.6) | 0.05 | 1.3 (1.0, 1.7) | 0.03 | 46.2 | 0.7 (0.3, 1.3) | 0.25 | 0.9 (0.4, 1.7) | 0.65 | -53.8 | 0.001 |
| 45+ | 80.0 | 1.0 (0.6, 1.7) | 0.91 | 1.0 (0.6, 1.8) | 0.89 | 50.0 | 0.7 (0.3, 2.1) | 0.54 | 0.9 (0.3, 2.6) | 0.86 | -30.0 | 0.081 |
| **Injection more than once per week** | | | | | | | | | | | | |
| No | 78.6 | 1 | 1 | 1 | 1 | 44.4 | 1 | | 1 | | -34.2 | 0.001 |
| Yes | 85.6 | 1.1 (0.9, 1.3) | 0.40 | 1.0 (0.9, 1.3) | 0.48 | 60.1 | 1.4 (0.8, 2.3) | 0.26 | 1.3 (0.7, 2.2) | 0.38 | -25.5 | 0.001 |
| **Drug treatment, last 12 months** | | | | | | | | | | | | |
| Yes | 89.2 | 1 | | 1 | | 50.8 | 1 | | 1 | | -38.4 | 0.001 |
| No | 81.5 | 0.9 (0.8, 1.0) | 0.06 | 0.9 (0.8, 1.0) | 0.06 | 62.0 | 1.2 (0.9, 1.6) | 0.15 | 1.3 (1.0, 1.6) | 0.09 | -19.5 | 0.001 |
| **Needle exchange, last 12 months** | | | | | | | | | | | | |
| Yes | 87.5 | 1 | | 1 | | 61.0 | 1 | | 1 | | -26.5 | 0.001 |
| No | 83.9 | 1.0 (0.9, 1.1) | 0.43 | 1.0 (0.9, 1.1) | 0.67 | 31.2 | 0.5 (0.2, 1.1) | 0.07 | 0.5 (0.3, 1.1) | 0.08 | -52.7 | 0.001 |

*Adjusted for age group, race/ethnicity, and gender.

San Francisco established a collective impact initiative in 2017 called "End Hep C SF" to eliminate HCV as a public health threat by 2030 [18]. The program focuses on four major efforts: 1) advocacy for policy and funding for better viral hepatitis response, 2) increasing mobile and fixed sites for HCV prevention, testing, and linkage to care, 3) improving access to treatment for PWID and other underserved populations, 4) and using research and surveillance data to define strategies and evaluate impact. Broad strategic planning influenced programmatic activities across San Francisco that increased the availability of HCV testing services and treatment opportunities beyond traditional clinical venues.

Following the availability of new direct-acting antiviral medications for HCV [6], results suggest HCV treatment prevalence among PWID in San Francisco has nearly tripled between 2015 and 2018. The improvement may be attributed to the activities facilitated under the End Hep C SF initiative's objective to increased screening, linkage, and treatment access and uptake among San Francisco residents, including PWID. For example, three new HCV testing and linkage programs in the city were funded that prioritized testing and linkage for people who were out of care and experiencing other financial hardships. Since 2016, community-based rapid-HCV testing and pre-planned linkage have provided HCV testing services to more than 2,000 individuals from marginalized communities. In 2016 alone, 578 patients engaged in medical care across the San Francisco Community Health Clinics, safety net clinics that serve many of those experiencing being homelessness, were treated for HCV [6]. This targeted effort is one possible reason we observed such an impressive decrease in the prevalence of untreated HCV infection among PWID recently homeless. Another program that reflects the overall strategy to reach PWID communities is an HCV test and treatment program rolled out in a methadone maintenance therapy (MMT) program and another within a harm reduction center, both showing to be cost-effective, acceptable and feasible [19]. Co-locating HCV test-and-

treat programs in venues frequented by PWID continues to be a productive strategy to achieve the goal of universal access for all patients to receive HCV treatment.

We found that in 2015 in San Francisco, only 47% of PWID had ever received an HCV diagnosis, of whom only 19% had ever received any medications for HCV. A similar proportion of PWID had ever received an HCV diagnosis in 2018 (50%), yet a much larger proportion had ever received HCV treatment (42%). Given the observed plateau in proportion of HCV diagnosis, more work needs to be done to increase HCV diagnostic testing and evaluation in this high-risk population.

Similar to previous studies [20–22], we found structural barriers for HCV treatment like unemployment and lack of health insurance may play a role. A significant factor identified was the lack of access to a usual source of health care. And while Medi-Cal (California's Medicaid program) removed many insurance related barriers to HCV DAAs medications, including removal of a sobriety requirement for drugs, not having medical insurance is likely a second indicator of being outside the medical service net [23]. Given the high prevalence of PWID without a usual source of health care, increasing community-based treatment programs co-located at services with social service programs that already serve PWID, like food pantries, could overcome this barrier.

In our adjusted sample, only 16% (n = 8) of young PWID (<30 years old) living with HCV had ever received medications for HCV. In a cohort study, only 17% (n = 5) of young PWID in San Francisco with an HCV diagnosis who referred to care had initiated HCV treatment [7]. Young PWID may face unique barriers to HCV treatment [22]. Perceived low consequences of HCV infection and misconceptions about HCV treatment availability for young people who are actively injecting drugs may result in significant barriers to uptake [20,22]. Those who access methadone maintenance therapy tend to be older. Perhaps developing youth-focused test-and-treat services may overcome some of the cultural barriers that keep young PWID from benefiting from HCV treatment programs co-located at MMT.

We recognize at least four major limitations of our study. First, we measured lifetime HCV testing, diagnosis, and treatment in the past by self-report (and not medical records), which is subject to misclassification. Second, we used a lifetime (ever) timeframe for our cascade analysis and our results do not differentiate treatment outcomes between PWID with old and recent HCV diagnosis. Third, survey questions did not ask participants what type of HCV medication they received, which did not allow us to differentiate the proportion who were treated with DAA versus older interferon therapy. Fourth, our results may not be generalizable beyond San Francisco given the high proportions of patients reporting health insurance and having accessed care.

To achieve the HCV elimination target by 2030 in San Francisco, more than 1,400 annual HCV treatments among PWID are required [24]. Expanding HCV case-finding and treatment programs, as well as scaling up prevention programs such as medication-assisted treatment and syringe service programs, is necessary to achieve the target by 2030. Integrated approaches to increase access to care are necessary to improve upon the HCV care cascade, particularly focusing on harder to reach PWID such as those who are young and those outside of regular care, ensuring these programs are sustainable to serve the needs of PWID when they are ready to begin HCV treatment.

## Supporting information

**S1 Fig. Crude estimates for the Hepatitis C virus (HCV) infection treatment cascade in two rounds of surveys of people who inject drugs (PWID) in San Francisco, 2015 and 2018.** Denominators for percent who received treatment and attained sustained virologic response (SVR) are those diagnosed with HCV infection. In 2015, tested antibody positive and

attained SVR were not measured. The estimates in this graph are not RDS-adjusted (i.e., crude). (DOCX)

## Acknowledgments

This work would not be possible without the study participants' willingness to provide their time and information.

## Author Contributions

**Conceptualization:** Ali Mirzazadeh, Katie Burk, Erin C. Wilson, Desmond Miller, Willi McFarland, Meghan D. Morris.

**Data curation:** Yea-Hung Chen, Jess Lin, Erin C. Wilson, Desmond Miller, Danielle Veloso, Willi McFarland.

**Formal analysis:** Ali Mirzazadeh, Yea-Hung Chen.

**Funding acquisition:** Erin C. Wilson, Willi McFarland.

**Investigation:** Jess Lin, Desmond Miller, Danielle Veloso, Willi McFarland.

**Methodology:** Ali Mirzazadeh, Yea-Hung Chen, Jess Lin, Katie Burk, Erin C. Wilson, Desmond Miller, Danielle Veloso, Willi McFarland, Meghan D. Morris.

**Project administration:** Danielle Veloso, Willi McFarland.

**Resources:** Erin C. Wilson, Willi McFarland.

**Software:** Yea-Hung Chen.

**Supervision:** Jess Lin, Desmond Miller, Willi McFarland, Meghan D. Morris.

**Visualization:** Ali Mirzazadeh.

**Writing – original draft:** Ali Mirzazadeh, Meghan D. Morris.

**Writing – review & editing:** Yea-Hung Chen, Jess Lin, Katie Burk, Erin C. Wilson, Desmond Miller, Danielle Veloso, Willi McFarland, Meghan D. Morris.

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
