## [Decision Letter · Decision Letter 0]

4 Jan 2021

PONE-D-20-31675

Progress toward closing gaps in the hepatitis C virus cascade of care for people who inject drugs in San Francisco

PLOS ONE

Dear Dr. Mirzazadeh,

Thank you for submitting your manuscript to PLOS ONE. After careful consideration, we feel that it has merit but does not fully meet PLOS ONE’s publication criteria as it currently stands. Therefore, we invite you to submit a revised version of the manuscript that addresses the points raised during the review process.

Your manuscript was reviewed by 2 experts in the field. Although both reviewers commented on high quality of your work, they identified important problems in your submission. Please consider the attached comments and provide point-by-point responses.

We look forward to receiving your revised manuscript.

Kind regards,

Yury E Khudyakov, PhD

Academic Editor

PLOS ONE

Journal Requirements:

"NO authors have competing interests"

Reviewers' comments:

Reviewer's Responses to Questions

**Comments to the Author**

1. Is the manuscript technically sound, and do the data support the conclusions?

Reviewer #1: Partly

Reviewer #2: Yes

2. Has the statistical analysis been performed appropriately and rigorously? 

Reviewer #1: Yes

Reviewer #2: Yes

3. Have the authors made all data underlying the findings in their manuscript fully available?

Reviewer #1: Yes

Reviewer #2: Yes

4. Is the manuscript presented in an intelligible fashion and written in standard English?

Reviewer #1: Yes

Reviewer #2: Yes

5. Review Comments to the Author

Reviewer #1: The manuscript reports on analyses from the NHBS IDU cycle data in San Francisco from 2015 and 2018 to ascertain trends in HCV treatment uptake. The authors report on important trends that may have implications for public health planning.

It is unclear how the self-reported “diagnosed with HCV infection” is defined and used. Was the participant ask if a physician diagnosed them? This would align with the reported majority of PWID in 2018 reporting visiting a physician in the prior 12 months (87.1%). However, the timing is unclear. Did these individuals got their HCV diagnosis from physician and were therefore more likely to be linked to care? Was it clear that it was chronic infection that required treatment? It is difficult to fully assess the impact of the increases in access to HCV treatment or link them to community efforts in San Francisco (End Hep C SF) without clarity on these points.

Minor:

The final paragraph is duplicated.

Figure 1 and Supplement Figure needs to indicate the difference in the data reported more clearly in the title (not just the y-axis), e.g., Crude vs. RDS-adjusted estimates

Reviewer #2: In their manuscript, the authors address a critically important issue in the continuum of care for Hepatitis C infection in patients that inject drugs. The manuscript is extremely well written with a clear presentation of the methods, data, and reasonable conclusions. The findings are very encouraging.

I had a few questions / comments:

1. In the continuum development, is there a distinction between "received a diagnosis" and a confirmed positive antibody result (positive HCV RNA)? Receiving a diagnosis and having the lab results to merit a diagnosis (but perhaps not having this adequately communicated to the patient) are slightly different, and likely are another pitfall in the care continuum. Did you all have access to the blood tests results? I think a sentence in the data analysis section to clarify this distinction may be helpful.

2. In the discussion, while I agree that the tripling of diagnosed patients receiving HCV therapy is fantastic news (and your method of showing this change is very effective), I'm not sure I'm as convinced about the stability of the infection rate from your data. I think the reference cited [20] is appropriate, but I think all you can really say in your data is that the proportion of respondents reporting a diagnosis of HCV (ever) is unchanged. I saw the stable diagnosis proportion as evidence that more work needs to be done to increase diagnostic testing and evaluation in this high risk population. I would be happy to be convinced otherwise.

3. The limitations are accurately described, but could use a bit more support. The first limitation combines the issues of self report and timing. I think these are different, and important limitations. For the timing, this is critical for the diagnosis portions of the continuum especially. I also think the comment about the generalizability of San Francisco is critical, given the remarkably high proportions of patients reporting health insurance and having accessed care.

4. The last paragraph of the discussion is nearly duplicated.

Overall, outstanding work.

6. PLOS authors have the option to publish the peer review history of their article (what does this mean?). If published, this will include your full peer review and any attached files.

Reviewer #1: No

Reviewer #2: No

---

## [Author Response · Author response to Decision Letter 0]

11 Feb 2021

5. Review Comments to the Author

Reviewer #1: The manuscript reports on analyses from the NHBS IDU cycle data in San Francisco from 2015 and 2018 to ascertain trends in HCV treatment uptake. The authors report on important trends that may have implications for public health planning.

It is unclear how the self-reported “diagnosed with HCV infection” is defined and used. Was the participant ask if a physician diagnosed them? This would align with the reported majority of PWID in 2018 reporting visiting a physician in the prior 12 months (87.1%). However, the timing is unclear. Did these individuals got their HCV diagnosis from physician and were therefore more likely to be linked to care? Was it clear that it was chronic infection that required treatment? It is difficult to fully assess the impact of the increases in access to HCV treatment or link them to community efforts in San Francisco (End Hep C SF) without clarity on these points.

** Response: we appreciate the positive feedback on the value of our paper’s message. As requested, we provided more details about how we measured the HCV testing, diagnosis and treatment in the survey. The timeframe for our analysis was “ever”.

Our analysis of cascade for HCV testing, diagnosis and treatment was based on the following survey questions: 

• Have you ever been tested for hepatitis C infection?

• Has a doctor, nurse or other health care provider ever told you that you had hepatitis C?

• Have you ever taken medicine to treat your hepatitis C infection?

Minor:

The final paragraph is duplicated.

** Response: We are sorry for this and removed the duplicate. 

Figure 1 and Supplement Figure needs to indicate the difference in the data reported more clearly in the title (not just the y-axis), e.g., Crude vs. RDS-adjusted estimates

** Response: we added details to the titles for Figure 1 and Supplement Figure 1 as suggested:

Fig 1. RDS-adjusted estimates for the Hepatitis C virus (HCV) infection treatment cascade in two rounds of surveys of people who inject drugs (PWID) in San Francisco, 2015 and 2018. Denominators for percent who received treatment and attained sustained virologic response (SVR) are those diagnosed with HCV infection. In 2015, tested antibody positive and attained SVR were not measured. The estimates in this graph are RDS-adjusted.

S1 Fig. Crude estimates for the Hepatitis C virus (HCV) infection treatment cascade in two rounds of surveys of people who inject drugs (PWID) in San Francisco, 2015 and 2018. Denominators for percent who received treatment and attained sustained virologic response (SVR) are those diagnosed with HCV infection. In 2015, tested antibody positive and attained SVR were not measured. The estimates in this graph are not RDS-adjusted (i.e., crude).

Reviewer #2: In their manuscript, the authors address a critically important issue in the continuum of care for Hepatitis C infection in patients that inject drugs. The manuscript is extremely well written with a clear presentation of the methods, data, and reasonable conclusions. The findings are very encouraging.

** Response: we appreciate the positive feedback on our paper.

I had a few questions / comments:

1. In the continuum development, is there a distinction between "received a diagnosis" and a confirmed positive antibody result (positive HCV RNA)? Receiving a diagnosis and having the lab results to merit a diagnosis (but perhaps not having this adequately communicated to the patient) are slightly different, and likely are another pitfall in the care continuum. Did you all have access to the blood tests results? I think a sentence in the data analysis section to clarify this distinction may be helpful.

** Response: you make a good point. By “received a diagnosis ”, we mean they “have been tested for hepatitis C infection in the past” and “a doctor, nurse or other health care provider ever told them that they had hepatitis C”. We made this clearer in the method section:

Our analysis of cascade for HCV testing, diagnosis and treatment was based on the following survey questions: 

• Have you ever been tested for hepatitis C infection?

• Has a doctor, nurse or other health care provider ever told you that you had hepatitis C?

• Have you ever taken medicine to treat your hepatitis C infection?

2. In the discussion, while I agree that the tripling of diagnosed patients receiving HCV therapy is fantastic news (and your method of showing this change is very effective), I'm not sure I'm as convinced about the stability of the infection rate from your data. I think the reference cited [20] is appropriate, but I think all you can really say in your data is that the proportion of respondents reporting a diagnosis of HCV (ever) is unchanged. I saw the stable diagnosis proportion as evidence that more work needs to be done to increase diagnostic testing and evaluation in this high risk population. I would be happy to be convinced otherwise.

** Response: We agree and revise the section to address your concern:

We found that in 2015 in San Francisco, only 47% of PWID had ever received an HCV diagnosis, of whom only 19% had ever received any medications for HCV. A similar proportion of PWID had ever received an HCV diagnosis in 2018 (50%), yet a much larger proportion had ever received HCV treatment (42%). Given the observed plateau in proportion of HCV diagnosis, more work needs to be done to increase HCV diagnostic testing and evaluation in this high-risk population.

3. The limitations are accurately described, but could use a bit more support. The first limitation combines the issues of self report and timing. I think these are different, and important limitations. For the timing, this is critical for the diagnosis portions of the continuum especially. I also think the comment about the generalizability of San Francisco is critical, given the remarkably high proportions of patients reporting health insurance and having accessed care.

** Response: We added more details to the limitation section as requested. 

We recognize at least four major limitations of our study. First, we measured lifetime HCV testing, diagnosis, and treatment in the past by self-report (and not medical records), which is subject to misclassification. Second, we used a lifetime (ever) timeframe for our cascade analysis and our results do not differentiate treatment outcomes between PWID with old and recent HCV diagnosis. Third, survey questions did not ask participants what type of HCV medication they received, which did not allow us to differentiate the proportion who were treated with DAA versus older interferon therapy. Fourth, our results may not be generalizable beyond San Francisco given the high proportions of patients reporting health insurance and having accessed care. 

4. The last paragraph of the discussion is nearly duplicated.

** Response: We are sorry for this and removed the duplicate. 

Overall, outstanding work.

** Response: Thank you very much.

---

## [Decision Letter · Decision Letter 1]

22 Mar 2021

Progress toward closing gaps in the hepatitis C virus cascade of care for people who inject drugs in San Francisco

PONE-D-20-31675R1

Dear Dr. Mirzazadeh,

We’re pleased to inform you that your manuscript has been judged scientifically suitable for publication and will be formally accepted for publication once it meets all outstanding technical requirements.

Kind regards,

Yury E Khudyakov, PhD

Academic Editor

PLOS ONE

Additional Editor Comments (optional):

Reviewers' comments:

Reviewer's Responses to Questions

**Comments to the Author**

1. If the authors have adequately addressed your comments raised in a previous round of review and you feel that this manuscript is now acceptable for publication, you may indicate that here to bypass the “Comments to the Author” section, enter your conflict of interest statement in the “Confidential to Editor” section, and submit your "Accept" recommendation.

Reviewer #1: All comments have been addressed

Reviewer #2: All comments have been addressed

2. Is the manuscript technically sound, and do the data support the conclusions?

Reviewer #1: Yes

Reviewer #2: Yes

3. Has the statistical analysis been performed appropriately and rigorously? 

Reviewer #1: Yes

Reviewer #2: Yes

4. Have the authors made all data underlying the findings in their manuscript fully available?

Reviewer #1: Yes

Reviewer #2: Yes

5. Is the manuscript presented in an intelligible fashion and written in standard English?

Reviewer #1: Yes

Reviewer #2: Yes

6. Review Comments to the Author

Reviewer #1: (No Response)

Reviewer #2: Thank you for addressing and expounding upon all questions / concerns. All of the responses are well integrated into the manuscript.

7. PLOS authors have the option to publish the peer review history of their article (what does this mean?). If published, this will include your full peer review and any attached files.

Reviewer #1: No

Reviewer #2: No

---

## [Editor Report · Acceptance letter]

24 Mar 2021

PONE-D-20-31675R1 

Progress toward closing gaps in the hepatitis C virus cascade of care for people who inject drugs in San Francisco 

Dear Dr. Mirzazadeh:

I'm pleased to inform you that your manuscript has been deemed suitable for publication in PLOS ONE. Congratulations! Your manuscript is now with our production department. 

Kind regards, 

on behalf of

Dr. Yury E Khudyakov 

Academic Editor

PLOS ONE